# Automatic synchronisation of the cell cycle in budding yeast through closed-loop feedback control

Giansimone Perrino [1,6], Sara Napolitano [1,2,6], Francesca Galdi [1], Antonella La Regina [1], Davide Fiore [3], Teresa Giuliano [1], Mario di Bernardo [4,5] & Diego di Bernardo [1,2 ✉]

The cell cycle is the process by which eukaryotic cells replicate. Yeast cells cycle asynchronously with each cell in the population budding at a different time. Although there are several experimental approaches to synchronise cells, these usually work only in the short-term. Here, we build a cyber-genetic system to achieve long-term synchronisation of the cell population, by interfacing genetically modified yeast cells with a computer by means of microfluidics to dynamically change medium, and a microscope to estimate cell cycle phases of individual cells. The computer implements a controller algorithm to decide when, and for how long, to change the growth medium to synchronise the cell-cycle across the population. Our work builds upon solid theoretical foundations provided by Control Engineering. In addition to providing an avenue for yeast cell cycle synchronisation, our work shows that control engineering can be used to automatically steer complex biological processes towards desired behaviours similarly to what is currently done with robots and autonomous vehicles.

[1] Telethon Institute of Genetics and Medicine (TIGEM), Pozzuoli, Italy. [2] Department of Chemical, Materials and Industrial Production Engineering, University of Naples Federico II, Naples, Italy. [3] Department of Mathematics and Applications "R. Caccioppoli", University of Naples Federico II, Naples, Italy. [4] Department of Electrical Engineering and Information Technology, University of Naples Federico II, Naples, Italy. [5] SSM - School for Advanced Studies, Naples, Italy. [6] These authors contributed equally: Giansimone Perrino, Sara Napolitano. ✉email: dibernardo@tigem.it

The cell cycle is the essential process by which eukaryotic cells replicate. It consists of a sequential series of events that are tightly controlled by an evolutionary conserved regulatory network leading to cell division[1]. In the budding yeast *S. cerevisiae*, the cell divides asymmetrically with a larger mother budding a smaller daughter cell as outlined in Fig. 1a[2,3]. The cell cycle can be divided into four phases: the growth phase ($G_1$), with a considerable increase in volume, which is followed by the DNA synthesis (S) phase, during which DNA is replicated; afterwards the cell enters a second growth phase ($G_2$), with the appearance of a bud that will grow into the daughter cell, and ends with the mitotic phase (M), when chromosomes become separated and cell division occurs, giving rise to the daughter cell.

During the $G_1$ phase, the cell can either commit to the cell cycle and enter the S phase, or if conditions are not favourable, arrest the cell cycle. In yeast, this START checkpoint is found in the late $G_1$ phase. Activation of the cyclin-dependent kinase (CDK) Cdc28 by any of the cyclins Cln1, Cln2 and Cln3 is necessary to overcome the START checkpoint[4–6] and to irreversibly enter the cell cycle.

Yeast cells cycle asynchronously, meaning that each cell in the population buds at a different time (Fig. 1b). Such desynchronised behaviour increases cellular heterogeneity and it may be advantageous in unicellular organisms for the survival of the population in unfavourable conditions[7,8]. There are cases, however, where a synchronised population is desirable. Indeed, the yeast *S. cerevisiae* is the model organism of choice to study the mechanisms underlying eukaryotic cell cycle regulation. To do so, however, scientists need a synchronised population of cells to enable robust measurements of morphological (e.g. cell size), phenotypical (e.g. cell-cycle duration) and molecular (e.g. protein expression) properties. Synchronised yeast cells can also benefit biotechnological applications to optimise cell-cycle modulated production of metabolites and heterologous proteins[9–11].

Several experimental approaches to synchronise cells[9,12–15] have been proposed over more than 20 years of research, and can be generally categorised into two broad classes, according to the underlying principles of physical selection or chemical induction. Physical selection methods rely on separating cells according to their size, which is related to the cell cycle phase, and then on growing cells with the same size together. In chemical induction methods, cells are blocked in the same cell-cycle phase by the presence, or absence, of one or more chemicals in the growth medium and then released altogether by replacing the growth medium. An alternative is to use cell division cycle mutants where the cell cycle can be induced by temperature shifts or changes in metabolite concentrations. All of these approaches, however, work only in the short-term, as cells are only forced to start in the same cell-cycle phase but then rapidly desynchronise after a few rounds of replication[16]. This quick desynchronisation happens in part because of asymmetry in the size of mother cells (larger) and daughter cells (smaller). Indeed, cells need to reach a critical volume before entering the cell cycle, thus limiting the duration of synchronisation[10]. Moreover, when cells are grown in secondary carbon sources (e.g. galactose), the cell cycle slows down thus enhancing the difference between mother and daughter cells and causing desynchronisation after only one cycle[10].

To solve this long-standing issue, continuous culture systems have also been proposed. These are based on modified chemostats where cells are periodically starved to block cell division and then exposed to nutrients; however, these systems are technically demanding, stressful for the cells and not robust, as the starvation pulses must be of a specific frequency and magnitude, which is not known beforehand[9,17]. A variation of this approach is the self-cycling fermenter where measurement of dissolved oxygen is used to detect the onset of nutrient depletion and trigger the removal of half the fermenter content and its replacement with

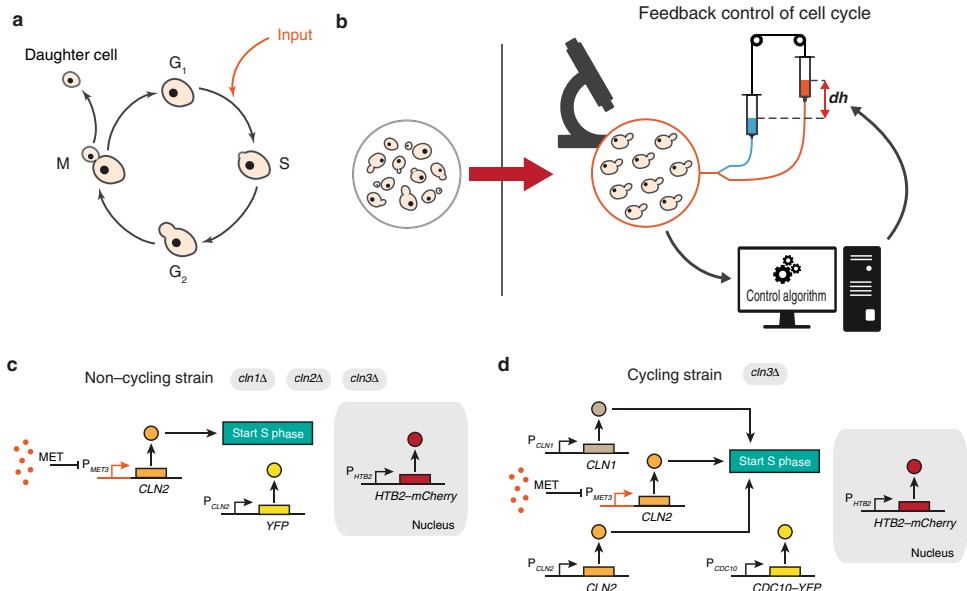

**Fig. 1 Synchronisation of cell cycle in yeast *S. cerevisiae* through automatic feedback control. a** Schematic representation of the cell cycle in yeast *Saccharomyces cerevisiae*. Yeast strains were engineered to initiate the cell cycle upon methionine depletion from the growth medium (input). **b** Yeast cells do not cycle synchronously in a population. Schematics of the computer-controlled microfluidics platform to automatically synchronise the cell cycle across a population of yeast cells. **c** Non-cycling yeast strain. Cells are deleted for genes encoding for the $G_1$ cyclins Cln1-3 while an exogenous $G_1$ cyclin gene *CLN2* is placed under the control of the methionine-repressible promoter $P_{MET3}$. Cells can cycle only in the absence of methionine. A yellow fluorescent protein (YFP) is expressed under the control of the endogenous promoter $P_{CLN2}$. A red fluorescent nuclear reporter, consisting of a fusion protein between the endogenous histone H2B protein and the mCherry (Htb2-mCherry) is also present in this strain. **d** Cycling yeast strain. Cells are deleted only for the $G_1$ cyclin Cln3, hence cells continuously cycle. An exogenous copy of the $G_1$ cyclin gene *CLN2* is placed under the control of the methionine-repressible promoter $P_{MET3}$. A red nuclear fluorescence reporter (Htb2-mCherry) and a yellow fluorescence reporter, consisting of a fusion protein between endogenous mitotic septin Cdc10 and YFP (Cdc10-YFP), are also present in this strain.

fresh medium[17]. These approaches, while easily scalable to large fermenters, require half of the cells to be discarded, which is not optimal, while causing cells to periodically exit the exponential growth phase and reach the stationary phase, thus stressing cells and slowing down cell growth.

Here, we ask whether we could design a robust approach to automatically keep the cell cycle indefinitely synchronised in a population of exponentially growing yeast cells independently of the carbon source and without requiring periodic nutrient depletion.

To tackle this problem, we turned to Control Engineering, a well-established discipline to build controllers to regulate the behaviour of physical systems reliably and robustly across a range of operating conditions. Closed-loop feedback control is the most common strategy used in Control Engineering to implement a controller. Feedback control relies on a sense and react paradigm (Fig. 1b), where the quantity to be regulated (e.g. the cell-cycle phase of each cell) is measured in real-time and then the controller, usually implemented in a computer, adjusts the input accordingly (e.g. duration and timing of the stimulation) to achieve the control objective (e.g. synchronisation of cell cycle across cells). A key theoretical result of control theory is that feedback control endows systems with robustness to perturbations and uncertainties. Applications of computer-based feedback control to biomolecular processes have appeared over the last ten years and rely on microscopy and image analysis to measure in real-time the quantity of interest (usually a fluorescent protein), and on optogenetics or microfluidics to deliver the control input[18–30]. Computer-based feedback control has been successfully applied to processes such as the regulation of gene expression from an inducible promoter, protein localisation, activity of endogenous pathways and neuronal activity[18–30].

Recently, Charvin et al.[31] were able to synchronise exponentially growing yeast cells in a microfluidics device by inducing periodic expression of the $G_1$ cyclin *CLN2* using a methionine-repressible promoter[32] in a cell division cycle mutant. Among their other findings, they observed that timing of methionine removal and administration (i.e. period and duration of the stimulation) must be carefully tuned to achieve satisfactory synchronisation. Moreover, changes in growth conditions, such as carbon source or temperature, will desynchronise the population, unless the optimal stimulation duration and timing is properly adjusted.

In this work, motivated by these findings, we develop a completely automated approach based on a microfluidics platform implementing a feedback control strategy able to maintain an exponentially growing yeast population synchronised over time despite changes in cellular and environmental conditions, as shown in Fig. 1b. Our approach opens the way to automatic control of yeast cell cycle, offering a tool to study the cell cycle and for biotechnological applications, while proving that complex biological processes can be regulated by computer-based feedback control[33].

## Results

**Yeast strains for inducible cell cycle start**. We made use of two different yeast strains genetically engineered to start the cell cycle upon removal of methionine from the growth medium as shown in Fig. 1c, d.

The first strain in Fig. 1c, which we referred to as the non-cycling strain, was engineered by Rahi et al.[34] (Methods). In the non-cycling strain, endogenous control of cell cycle initiation is disrupted by deletion of the genes encoding for the $G_1$ cyclins Cln1, Cln2 and Cln3[35], whereas *CLN2* is placed downstream of the methionine-repressible promoter $P_{MET3}$ to allow its inducible

expression[32]. A yellow fluorescence protein (YFP) is expressed from the endogenous $P_{CLN2}$ promoter and thus peaks in the late $G_1$ phase (Methods). Finally, a constitutively expressed histone Htb2-mCherry acts as a nuclear fluorescence marker for facilitating image analysis (Methods). The non-cycling strain is blocked in the $G_1$ phase when grown in methionine-rich medium.

The second strain, which we call the cycling strain, is shown in Fig. 1d and it was derived from the one described by Charvin et al.[31]. In this strain, endogenous control of cell cycle initiation is maintained by preserving the genes encoding the $G_1$ cyclins Cln1 and Cln2 and by deleting Cln3[35]. An extra copy of *CLN2* is placed under the control of the methionine-repressible promoter $P_{MET3}$. Two fluorescence markers are present in this strain: the mitotic septin Cdc10-YFP, expressed during the S-$G_2$-M phases, and the Htb2-mCherry as a constitutive nuclear marker (Methods). The cycling strain can cycle independently of methionine levels in the growth medium, however, cells can be forced to transition from $G_1$ to the S phase on demand by inducing exogeneous *CLN2* expression via methionine removal.

The rationale of using two different strains is that cell-cycle synchronisation in the cycling strain is intrinsically much more challenging than in the non-cycling strain, as the cells continuously cycle and can never be blocked. As an analogy, imagine a group of basketball players, each constantly moving while bouncing a ball, the challenge is to synchronise the players so that all the balls touch the ground at the same time. If the players can hold the ball still (non-cycling strain) than it suffices to periodically ask the players to hold the ball and release it at the same time (stop&go strategy described below); however, if the players are not allowed to hold the ball (cycling strain) then the task becomes much more difficult.

**Automatic feedback control of cell cycle in the non-cycling yeast strain**. We first experimentally characterised the non-cycling strain. To this end, we grew cells in a microfluidics platform to dynamically change the micro-environment, while measuring in real-time fluorescence reporters in individual cells by microscopy, as shown in Fig. 1b (Methods). In methionine-rich medium, cells grow in volume but fail to divide for at least 6 h (Supplementary Fig. 1a–c, g); indeed, some cells were able to replicate after this time, albeit very slowly, possibly because of accumulation of *CLN2* caused by promoter's leakiness.

In methionine-depleted medium (Fig. 2a–e), cells are able to cycle as evidenced by the exponential increase in cell number over time (Fig. 2a and Supplementary Fig. 1g) and by the cyclic expression of the YFP reporter in individual cells (Fig. 2c, Supplementary Fig. 1d–f and Supplementary Movie 1). Interestingly, the YFP fluorescence intensity averaged across the population shows a flat profile (Fig. 2b), despite being oscillatory in individual cells. This behaviour stems from cells not cycling in phase, thus causing individual oscillations in fluorescence to cancel out when computing the average intensity. Finally, we estimated the budding index (B.I.) from the YFP fluorescence intensities (Methods). The B.I. represents the percentage of cells in the budded phase (i.e. S-$G_2$-M phases) at each time point (Fig. 2d) and it is expected to remain constant over time for an unsynchronised cell population. Our experimental characterisation of the non-cycling yeast strain thus confirmed that it behaves as originally described[34].

Next, we asked whether periodic expression of the $G_1$ cyclin *CLN2* in response to periodic pulses of methionine depletion (−MET) would cause individual cells to cycle synchronously. To choose the period ($T_u$) and duration ($D_{-Met}$) of such pulses, we first derived and analysed a deterministic mathematical model of the cell cycle (Methods). Briefly, the cell cycle was modelled as a

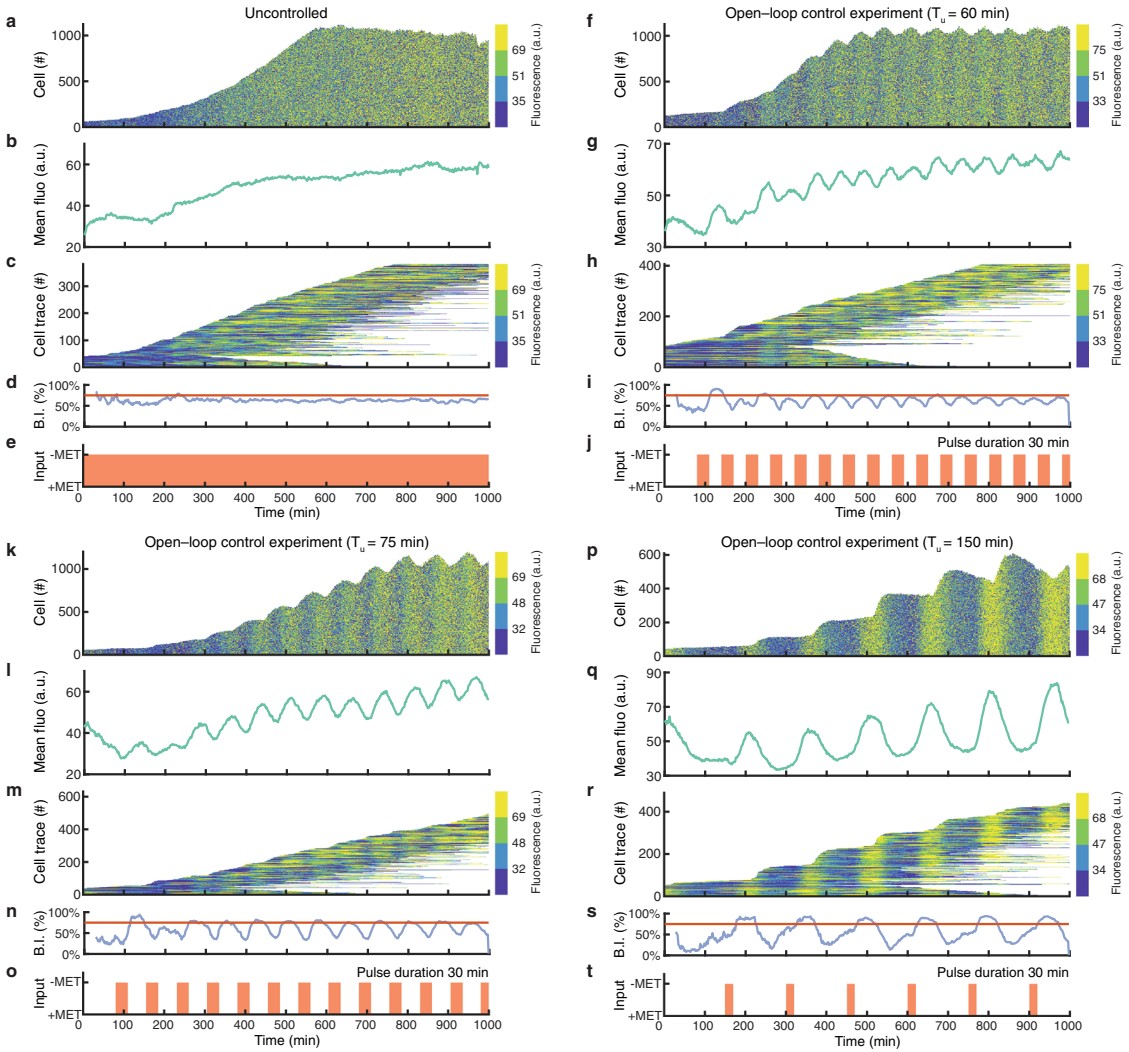

**Fig. 2 Characterisation and open-loop control of the non-cycling yeast strain.** Experiments with non-cycling yeast strain cells grown in the automated microfluidics platform in four different conditions: (**a–e**) methionine-depleted medium (−MET); (**f–j**) alternating pulses of methionine-rich (+MET) and methionine-depleted (−MET) medium with a period $T_u = 60$ min; (**k–o**) alternating pulses with a period $T_u = 75$ min; (**p–t**) alternating pulses with a period $T_u = 150$. Duration of −MET pulse was set to 30 min. **a**, **f**, **k**, **p** The number of cells and the distribution of YFP fluorescence intensity in the population over time. Fluorescence values are binned into 4 colours, corresponding to the quartiles, for clarity of visualisation. **b**, **g**, **l**, **q** Average YFP fluorescence intensity in the cell population. **c**, **h**, **m**, **r** Single-cell fluorescence traces over time. Each horizontal line corresponds to one cell. Each line starts when the cell is first detected and ends when the cell exits the field of view. The number of tracked cells does not correspond to the total number of cells as only cells tracked for longer than 300 min are shown. **d**, **i**, **n**, **s** Budding index (blue) reporting the percentage of cells in the budding phase (S-G₂-M) computed from the estimated cell cycle phases. The red line denotes the expected value of the budding index in the case of a totally desynchronised cell population. **e**, **j**, **o**, **t** Growth medium delivered to the cells as a function of time: +MET methionine-rich medium, −MET: methionine-depleted medium.

phase oscillator, which can be depicted as a clock with a single moving arm whose position indicates the phase of the cell and whose length is proportional to the cell volume (Supplementary Fig. 2a). To simplify the model, we assumed that volume growth in the mother cell occurs only in the $G_1$ phase, whereas the bud grows in volume only during the S-G₂-M phases. In the model, the cell will stop in the $G_1$ phase as long as methionine is present in the medium and will jump to the S phase in its absence, but only if the cell volume is above a critical threshold. Cell cycle duration in the absence of methionine was set to 75 min based on the literature[36]. To simulate a growing population of yeast cells, we generated an agent-based model where each cell is a phase oscillator, and a new agent is added after each cell cycle is completed. We first simulated the mathematical model to predict the behaviour of the cell population in response to external periodic expression of *CLN2* by varying the stimulation period $T_u$

and the pulse duration $D_{-Met}$ (Supplementary Fig. 2b, d). Numerical simulations show that the forcing period $T_u$ is of paramount importance for achieving cell cycle synchronisation across the population. Indeed, to fully synchronise the population, the period $T_u$ must be greater than the intrinsic cell cycle period (i.e. $T_u \geq 75$ min) (Supplementary Fig. 2b). Moreover, the longer the period ($T_u$), the larger the average cell volume ($V$) (Supplementary Fig. 2d), as cells stay in the $G_1$ phase for longer.

Guided by these numerical results, we performed microfluidics-based experiments to periodically induce the expression of *CLN2* with −MET pulses of period $T_u$ varying between 60 min and 150 min, and duration $D_{-Met} = 20$ min or 30 min. Experimental results are consistent with numerical simulations, as both synchronisation of the cell cycle across the population and the cells' average volume increase with the period $T_u$ (Fig. 2f–t, Supplementary Fig. 3 and Supplementary Movies 2

and 3). The fluorescence intensities of cells over time, in Fig. 2f, k and p, show a clear vertical pattern, indicating that cells are mostly in the same cell cycle phase; additionally, the number of cells increases in a stepwise fashion, rather than exponentially, as most of the cells bud together. Furthermore, the population averaged YFP fluorescence intensity (Fig. 2g, l, q) displays an oscillatory behaviour, as cells become synchronised. A similar result was obtained by computing the budding index (Fig. 2i, n, s) from single-cell traces (Fig. 2h, m, r). Finally, we observed that for a shorter duration of −MET pulses ($D_{-Met} = 20$ min and $T_u = 75$ min), cells' synchronisation takes considerably longer (700 min) as compared to longer pulses ($D_{-Met} = 30$ min and $T_u = 75$ min), (Supplementary Fig. 3 and Fig. 2k–o). The extent of synchronisation was quantified for each experiment and reported in Supplementary Fig. 4a, c, e. Specifically, we measured: the time-average of the mean phase coherence ($R$), a statistical measure of phase synchronisation estimated from the cell-cycle phase of individual cells (Methods); the amplitude (Power) and period of the leading peak of the power spectrum of population averaged YFP fluorescence signal (Methods). Briefly, the higher the values of $R$ and Power, the better the synchronisation. Quantitative analysis of the experiments in Supplementary Fig. 4a, c, e confirmed that it is possible to synchronise the cell cycle across the population by periodic stimulation, and that synchronisation increases with the stimulation period $T_u$ ($R = 0.28$ and Power $= 36.33$ dB when $T_u = 60$ min; $R = 0.38$ and Power $= 44.50$ dB when $T_u = 75$ min; $R = 0.46$ and Power $= 41.29$ dB when $T_u = 150$ min). For completeness, we also estimated the time-average of the mean cell radius in Supplementary Fig. 4g.

Notwithstanding the feasibility of synchronising cells using periodic stimulation, this open-loop control strategy is highly sensitive to environmental conditions, for example, whereas periodic stimulation with period $T_u = 75$ min and pulse-duration $D_{-Met} = 30$ min is able to synchronise the cell cycle when cells are grown in nominal conditions (glucose at 30 °C in Fig. 2k–o; $R = 0.38$ and Power $= 44.52$ dB; Supplementary Fig. 4a, c, e), this is not the case when growing cells in galactose at 30 °C (Fig. 3k–o; $R = 0.21$ and Power $= 28.27$ dB, when $T_u = 75$ min and $D_{-Met} = 30$ min; Supplementary Fig. 4a, c, e) or in glucose at 27 °C (Supplementary Fig. 5a–e; $R = 0.28$ and Power $= 36.32$ dB, when $T_u = 75$ min and $D_{-Met} = 30$ min; Supplementary Fig. 4a, c, e).

To overcome the drawbacks of open-loop control, we implemented a closed-loop feedback control strategy to automatically synchronise the cell population. To this end, we exploited the characteristics of the non-cycling strain to be blocked in the $G_1$ phase in methionine-rich medium (+MET) and to cycle upon its depletion (−MET). We devised an event-triggered stop&go control strategy (Methods), whose working principle is illustrated in Fig. 3a: at each sampling time, the controller estimates the cell cycle phase of each cell across the population from fluorescence microscopy images (Methods). If the percentage of cells in the $G_1$ phase is higher than a fixed threshold, then the controller delivers an exogenous pulse of −MET, thus enabling the cell cycle to start in all $G_1$ phase cells.

To test the feasibility of the stop&go control strategy in achieving the synchronisation task, we first carried out a numerical simulation with a threshold value set to 50% (Fig. 3b–e). Numerical results show that the stop&go controller can in principle synchronise the cell cycle across a population of cells, without any prior knowledge of the cell cycle duration and growth conditions. Figure 3b shows the simulated fluorescence signal in individual cells over time, with clear vertical lines and a stepwise increase in cell number indicating synchronisation; the population average fluorescence signal (Fig. 3c) and the budding index (Fig. 3d) are oscillatory, as expected for synchronised cells.

Interestingly, after a transient, the external input became a quasi-periodic signal (Fig. 3e) and entrained the single-cell traces with a 1:1 ratio (Fig. 3b). Finally, we carried out numerical simulations for different values of the threshold used by stop&go controller to decide when to deliver the −MET pulse. As shown in Supplementary Fig. 2c, e, the larger the threshold, the more the cells become synchronised, but at the cost of larger cell volumes.

Encouraged by the numerical results, we experimentally implemented the stop&go feedback control strategy with a threshold set to 50%. Cells were grown overnight in the absence of methionine so that, at the beginning of each experiment, the population was totally desynchronised (Methods). An initial calibration phase of 30 min was required to initialise the phase estimation algorithm (Methods); after that the stop&go controller was activated for 500 min and then deactivated. The closed-loop stop&go control strategy was able to automatically synchronise the cell population (Fig. 3f–j: $R = 0.42$ and Power $= 40.52$ dB; with replicate experiments in Supplementary Fig. 6): fluorescence intensities including all of the cells (Fig. 3f; Supplementary Fig. 6a, f and Supplementary Movie 4) and single-cell fluorescence traces, available only for a subset of cells for which cell tracking, (Fig. 3h; Supplementary Fig. 6c, h) show a clear vertical pattern over time, indicating synchronisation, while cell number increases in a stepwise fashion, as long as the controller is active. In addition, the average fluorescence signal (Fig. 3g; Supplementary Fig. 6b, g) and the budding index (Fig. 3i; Supplementary Fig. 6d, i) show an oscillatory behaviour. As observed in simulations, after a transient, the control input became a quasi-periodic signal (Fig. 3j: Period equals to 95 min; Supplementary Fig. 6e, j). Quantification of the extent of synchronisation and average cell radius are reported for all the experiments in Supplementary Fig. 4a, c, e, g.

To check the robustness of the stop&go feedback controller and to compare it to the open-loop control strategy, we performed additional perturbation experiments by either changing the growth temperature or the carbon source. When the cell growth temperature was set to 27 °C (Supplementary Fig. 5), the closed-loop controller was still able to synchronise the cell population (Supplementary Fig. 5f–j) with a similar performance as in the unperturbed case ($R = 0.39$ and Power $= 47.44$ dB; Supplementary Fig. 4a, c, e), unlike the open-loop periodic stimulation strategy ($R = 0.28$ and Power $= 36.32$ dB; Supplementary Fig. 4a, c, e). Next, we performed closed-loop control with cells growing in galactose, which is known to slow down the cell cycle[37]. As shown in Fig. 3k–t, and quantified in Supplementary Fig. 4a, c, e, the closed-loop feedback control achieved synchronisation without any drop in performance ($R = 0.41$ and Power $= 40.39$ dB; Supplementary Fig. 4a, c, e), whereas open-loop periodic stimulation failed to synchronise the population ($R = 0.21$ and Power $= 28.27$ dB; Supplementary Fig. 4a, c, e).

Taken together, our results demonstrate that closed-loop feedback control can automatically synchronise yeast cells over time, even in the face of new and unexpected environmental perturbations.

**Automatic feedback control of cell cycle in a cycling yeast strain.** Next, we investigated cell cycle synchronisation in the cycling yeast strain in Fig. 1d. The main difference with respect to the non-cycling strain is that cells can cycle also in the presence of methionine. Nevertheless, when in the $G_1$ phase, cells can be forced to transition to the S phase by removing methionine from the medium.

We first performed a series of time-lapse microfluidics experiments to assess cell cycle dynamics both in the absence

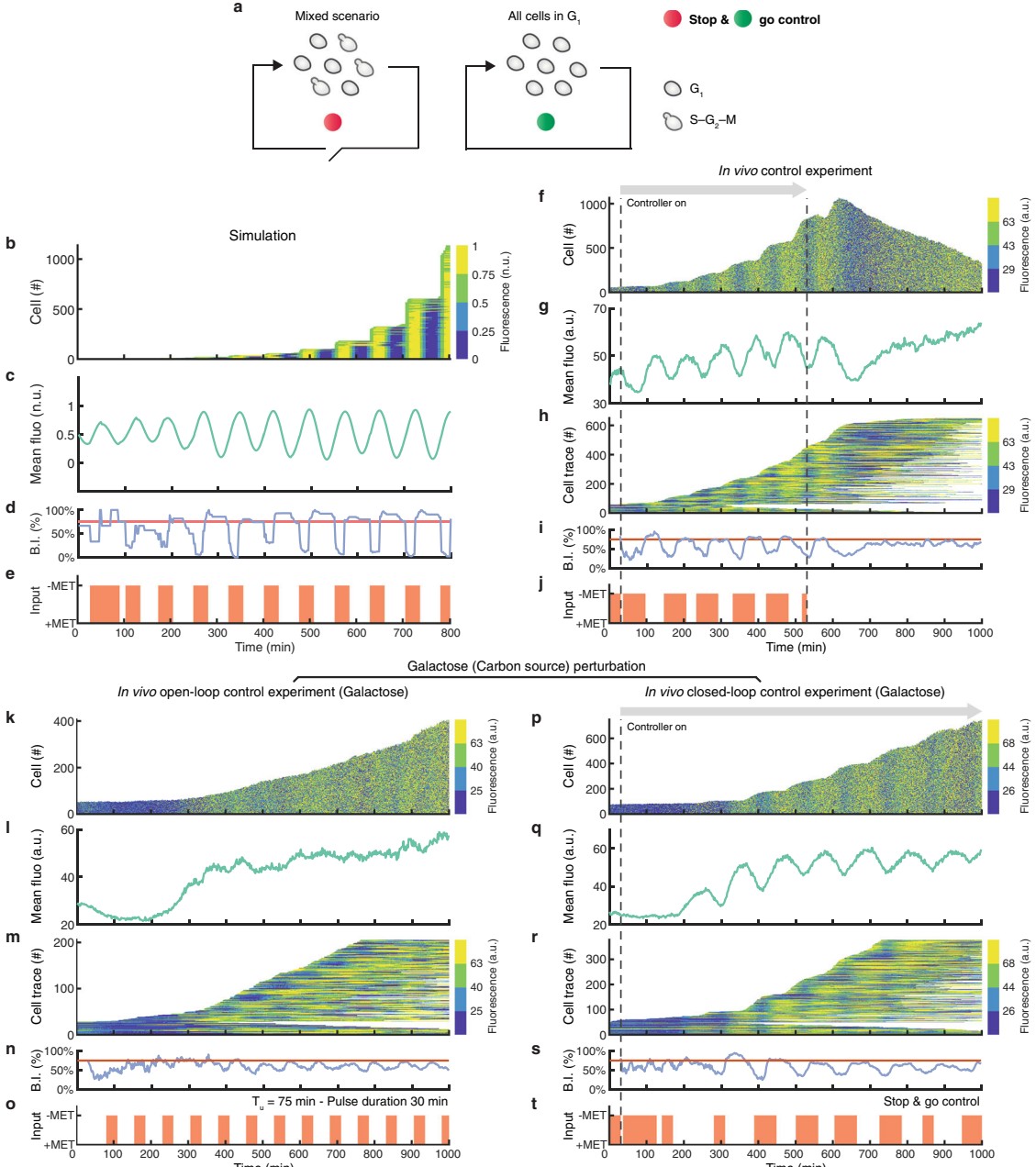

**Fig. 3 Automatic feedback control enables cell cycle synchronisation in the non-cycling yeast strain. a** Depiction of the stop&go control strategy. The controller waits for cells to stop in the $G_1$ phase and then restarts their cell cycle by removing methionine from the medium. **b–e**, Numerical simulation of the stop&go control strategy. **f–j**, Experimental implementation of the stop&go control strategy. An initial calibration phase of 30 min was required to set up the phase estimation algorithm. Dashed lines indicate the start and the end of the control experiment, after which cells are grown in methionine-rich medium. **k–o**, Open-loop experiment subjecting cells to alternating pulses of methionine-rich (+MET) and methionine-depleted (−MET) medium with a period $T_u = 75$ min using galactose (2% w/v) as carbon source. **p–t**, Experimental implementation of the stop&go control strategy using galactose (2% w/v) as carbon source. **b**, **f**, **k**, **p** The number of cells and the distribution of YFP fluorescence intensity in the population over time. Fluorescence values are binned into 4 colours, corresponding to the quartiles, for clarity of visualisation. **c**, **g**, **l**, **q** Average YFP fluorescence intensity in the cell population. **h**, **m**, **r** Single-cell fluorescence traces over time. Each horizontal line corresponds to one cell. Each line starts when the cell is first detected and ends when the cell exits the field of view. The number of tracked cells does not correspond to the total number of cells as only cells tracked for longer than 300 min are shown. **d**, **i**, **n**, **s** Budding index (blue) reporting the percentage of cells in the budding phase (S-$G_2$-M) computed from the estimated cell cycle phases. The red line denotes the expected value of the budding index in the case of a totally desynchronised cell population. **e**, **j**, **o**, **t** Growth medium delivered to the cells as a function of time: +MET methionine-rich medium, −MET: methionine-depleted medium.

(Supplementary Fig. 7a–e) and in the presence of methionine (Supplementary Fig. 7f–j and Supplementary Movie 5). As expected in both conditions, the number of cells increases exponentially over time (Supplementary Fig. 7a,f and Supplementary Movie 5) while cells are desynchronised as evidenced by

the flat profile in the average YFP fluorescence intensity (Supplementary Fig. 7b, g) and in the budding index (Supplementary Fig. 7d, i), and by quantitative analysis (Supplementary Fig. 4b, d, f). Charvin et al. have previously demonstrated that, in this strain, periodic expression of cyclin *CLN2* by alternating

pulses of methionine depletion can induce synchronisation of cell cycle across the population[31]. They also showed that synchronisation is very sensitive to the period of stimulation[31], achieving the best performance when cells are periodically stimulated with −MET pulses of period $T_u = 80$ min and duration $D_{-Met} = 20$ min[31]. To check the extent of synchronisation using this optimal open-loop control strategy in our experimental setup, we performed a microfluidics-based experiment to periodically induce the expression of *CLN2* with −MET pulses of period $T_u = 80$ min and duration $D_{-Met} = 20$ min (Supplementary Fig. 7k–o). Experimental results confirmed the synchronisation of cell cycle across the population ($R = 0.31$ and Power = 32.54 dB in Supplementary Fig. 4b, d, f).

We then asked whether a closed-loop feedback control strategy could improve synchronisation performance and robustness in this strain. The stop&go strategy is obviously not applicable, we thus implemented a reference oscillator feedback control strategy, which was adapted from the one proposed by Bai and Wen to synchronise a homogenous population of nonlinear phase oscillators using a common input[38]. At the core of this strategy lies a continuously cycling computer-generated in silico virtual yeast cell coupled to the real cells via the microfluidics device according to a star-like topology (Fig. 4a). In this context, the virtual cell behaves as a reference oscillator for all the other cells. Using the basketball players analogy, this is like projecting a virtual player on the wall and asking the real players to bounce the balls in synchrony with the virtual player. Mathematical details of this strategy are described in the "Methods" section.

To test the feasibility of reference oscillator feedback control strategy in achieving the synchronisation task, we first carried out a numerical simulation using an agent-based mathematical model (Methods). Numerical simulations of the reference oscillator strategy in Fig. 4b–e show effective synchronisation of the cell cycle population.

We thus tested the reference oscillator feedback control strategy experimentally and assessed its synchronisation performance in three separate experiments, as reported in Fig. 4f–j, Supplementary Fig. 8a–j and Supplementary Movie 6. In each experiment, cells were grown overnight in methionine-rich medium and thus were totally desynchronised at the beginning of the experiment (Methods). An initial calibration phase of 100 min was required to set up the phase estimation algorithm (Methods). We tested the reference oscillator strategy by performing a closed-loop control experiment for 600 min (Fig. 4f–j) and two additional ones for 500 min (Supplementary Fig. 8a–j). The controller was able to quickly synchronise the cell population in agreement with numerical simulations, as evidenced by the large oscillations in the average YFP fluorescence intensity and in the budding index in Fig. 4g, i and Supplementary Fig. 8b, d, g, i; and by the quantitative analysis ($R = 0.40$ and Power = 36.24 dB for the experiment in Fig. 4f–j; Supplementary Fig. 4b, d, f). We also confirmed that once the controller is turned off, cells quickly desynchronise (Fig. 4f–j, Supplementary Fig. 8a–j). To further assess the performance of the reference oscillator control strategy, we checked how well the cell-cycle phase of yeast cells in the population tracks one of the in silico reference oscillator. Specifically, we computed Spearman's rank correlation coefficient $\rho$ between the reference oscillator phase $\vartheta_r$ and the average phase $\psi$ of the cell population for each of the three reference control experiments ($\rho = [0.84, 0.81, 0.95]$, Supplementary Fig. 9a–c). Taken together, these results demonstrate the feasibility of synchronising the cell cycle across the population using the reference oscillator control strategy. Moreover, quantitative comparison of the performances of the open- and the closed-loop control strategies in Supplementary Fig. 4b, d, f confirms the superiority of the latter.

To check the robustness of the reference oscillator feedback controller, we performed an additional experiment by changing the carbon source from glucose to galactose. When cells were grown in galactose (Fig. 4k–t), the reference oscillator controller was still able to synchronise the cell population (Fig. 4p–t) with only a slight drop in performance ($R = 0.32$, Power = 25.17 dB and $\rho = 0.86$; Supplementary Fig. 4b, d, f and Supplementary Fig. 9d). On the other hand, the open-loop control strategy failed to satisfactorily synchronise the population as shown in Fig. 4k–o and in the quantitative analysis ($R$: 0.28; Power: N.D.; Supplementary Fig. 4b, d, f). These results demonstrate that the open-loop control strategy is highly sensitive to environmental conditions, whereas the closed-loop control strategy is robust.

## Discussion

A defining feature of yeast is its ability to bud daughter cells by following a precise sequence of events. Each cell within a population, however, buds at a different time (unsynchronised). Biologists and biotechnologists have long searched for methods to synchronise cells, but only short-term synchronisation has been achieved so far[16]. Here, we built a completely automatic system able to achieve long-term synchronisation of the cell population, by interfacing genetically modified yeast cells with a computer by means of microfluidics to dynamically change medium, and a microscope to estimate cell cycle phases of individual cells. The computer implements a controller algorithm to decide when, and for how long, to change the growth medium to synchronise the cell cycle across the population. Our work builds upon solid theoretical foundations provided by Control Engineering. This blending of disciplines is at the core of the new field of Cyber-genetics that aims at augmenting biological systems with controllers to regulate their behaviour for biotechnological and biomedical applications[22]. In addition to providing an avenue for yeast cell cycle synchronisation, our work shows that classical control engineering, originally devised for steering the behaviour of physical systems, can be successfully applied also to complex biological processes such as the cell cycle. Indeed, to the best of our knowledge, this is the first demonstration of computer-based feedback control of an endogenous system as complex as *S. cerevisiae*'s cell cycle. Our approach integrates microfluidics, microscopy, and image analysis to measure in real-time the level of synchronisation of the cell population and to apply the appropriate control input to achieve synchronisation. The robustness intrinsic to closed-loop feedback control enables synchronisation to automatically occur in a variety of environmental conditions (i.e. carbon source, temperature), unlike open-loop approaches, which require to preset the period and duration of the stimulation and are very sensitive to environmental fluctuations. Closed-loop approaches are more complex to implement than open-loop approaches, especially in the industrial setting. However, our approach can be easily scaled up from microfluidics to chemostats by switching to optogenetics-based gene expression control systems[26,27,33], thus opening the way for its use also in industrial applications. In addition, our approach could also be used to facilitate open-loop control strategies as it can automatically identify the best period and duration of stimulation to apply to yeast strains of scientific and commercial interest in different growth media and environmental conditions.

## Methods

**Yeast strains and growth conditions**. All *Saccharomyces cerevisiae* strains used in this study are listed in Supplementary Table 1. Strains are congenic with W303 strain. The SJR14a4d strain[34] (a kind gift from S. J. Rahi) is the non-cycling strain. The yDdB028 strain was derived from the GC84-35B strain[31] (a kind gift from G. Charvin) and represents the cycling strain.

The yDdB028 strain was constructed using standard procedures. The *HTB2-mCherry* cassette was amplified via PCR on genomic DNA extracted from the

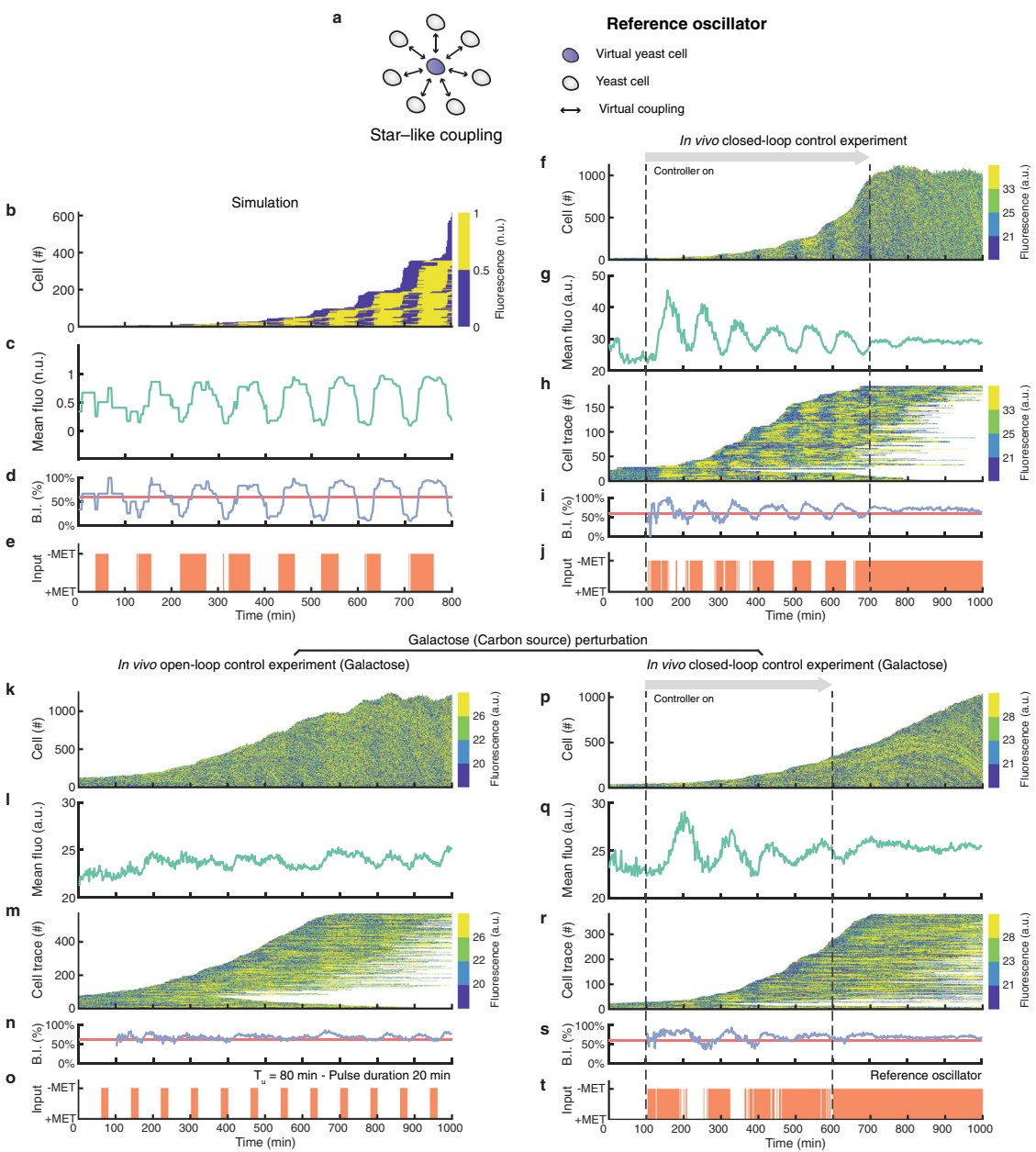

**Fig. 4 Automatic feedback control enables cell cycle synchronisation of the cycling yeast strain. a** Schematic illustration of the reference oscillator control strategy. Yeast cells are coupled to a virtual cell so that each cell cycles in sync with the virtual cell. **b–e** Numerical simulation of the reference oscillator control strategy. **f–j** Experimental implementation of the reference oscillator control strategy. An initial calibration phase of 100 min is required to set up the phase estimation algorithm. Dashed lines indicate the start and the end of the control experiment, after which cells are grown in methionine-depleted medium. **k–o** Open-loop experiment subjecting cells to alternating pulses of methionine-rich (+MET) and methionine-depleted (−MET) medium with a period $T_u = 80$ min using galactose (2% w/v) as carbon source. **p–t** Experimental implementation of the reference oscillator control strategy using galactose (2% w/v) as carbon source. **b, f, k, p** The number of cells and the distribution of YFP fluorescence intensity in the population over time. Fluorescence values are binned into 4 colours, corresponding to the quartiles, for clarity of visualisation. **c, g, l, q** Average YFP fluorescence intensity in the cell population. **h, m, r** Single-cell fluorescence traces over time. Each horizontal line corresponds to one cell. Each line starts when the cell is first detected and ends when the cell exits the field of view. The number of tracked cells does not correspond to the total number of cells as only cells tracked for longer than 300 min are shown. **d, i, n, s** Budding index (blue) reporting the percentage of cells in the budding phase (S-G$_2$-M) computed from estimated cell cycle phases. Red line denotes the expected value of the budding index in the case of a totally desynchronised cell population. **e, j, o, t** Growth medium delivered to the cells as a function of time: +MET methionine-rich medium, −MET: methionine-depleted medium.

SJR14a4d strain and cloned into plasmid pRS41N-GAP-CYC (a derivative of the nourseothricin-selective pRS41N plasmid[39] containing the CYC terminator, a kind gift from C. Wilson). The plasmid sequence was checked by Sanger sequencing. The *HTB2-mCherry* cassette was amplified from plasmid pRS41N-GAP-HTB2-mCherry-CYC and inserted into the yeast HTB2 locus via homologous recombination[40] such that expression was driven by the endogenous promoter. Correct integration was verified by PCR on extracted genomic DNA. All plasmids

used in this study are listed in Supplementary Table 2. All primers used in this study are listed in Supplementary Table 3.

Unless otherwise specified, both strains were grown at 30 °C in either synthetic complete medium, composed of yeast nitrogen base (0.67% w/v) with all amino acids; or synthetic complete drop-out medium, composed of yeast nitrogen base (0.67% w/v) with all amino acids except methionine; both supplemented with glucose (2% w/v) as carbon source. For the carbon source

perturbation experiments, synthetic complete media were supplemented with galactose (2% w/v).

**Microfluidics**. All microfluidics experiments were performed with the MFD0005a device[41]. This device contains a chamber in which the cells are trapped. The height of the chamber (3.5 μm) allows the yeast cells to grow only in a monolayer, simplifying the image analysis. Microfluidics devices were fabricated with a replica moulding technique as previously described[19]. Briefly, the master-mould was produced using multilayer soft-lithography with SU-8 as photoresist, and exposed to vapours of chlorotrimethylsilane (CARLO ERBA Reagents) for 10 min. Poly-dimethylsiloxane (PDMS; Sylgard 184, Dow Corning) was poured on the top of the master-mould with a 1:10 ratio (curing agent to base; w/w) and cured at 80 °C for 2 hrs. Next, the PDMS layer was cut and peeled from the master-mould and inlet ports were pierced with a micro-puncher (0.5 mm; World Precision Instruments) and then irreversibly bonded to a cover glass (thickness no. 1.5; Marienfeld Superior) through a plasma treatment for 30 s in a low-pressure plasma machine (ZEPTO version B; Diener electronic). The bonded device was baked for 2 hrs at 80 °C, and then stored at 4 °C until use. The fluid that reaches the chamber of the microfluidics device is a mixture of the growth media coming from the two inlet ports. The blending of the growth media depends on the relative pressure between the two fluids at the inlet ports. In order to change the relative pressure, we devised an automated actuation system that varies the relative height of the two syringes filled with the +MET and −MET media. The actuation system relies on two custom vertically mounted linear actuators, that can move independently. Each of the two linear actuators comprises one stepper motor connected with a syringe through a timing belt and two pulleys. A custom MATLAB script pilots the stepper motors to automatically move the syringes and thus direct the fluid from one or the other inlet to the cell chamber.

**Microscopy image acquisition and processing**. Phase contrast and epifluorescence images were acquired at 2 min intervals at ×40 magnification (CFI Plan Fluor DLL 40× dry objective, NA 0.75; Nikon Instruments) using a Nikon Eclipse Ti-E inverted microscope (Nikon Instruments) coupled with an EMCCD cooled camera (iXon Ultra 897; Andor Technology). The microscope stage was surrounded by an opaque cage incubator (Okolab) able to maintain the temperature at either 30 °C (nominal condition) or 27 °C (perturbed condition). Time-lapse experiments were conducted with the Perfect Focus System (Nikon Instruments) enabled. Appropriate filter cubes were used to acquire the yellow (YFP HYQ and FITC for the cycling and the non-cycling strains, respectively; Nikon Instruments) and the red (TRITC HYQ; Nikon Instruments) fluorescence channels. Time-lapse image acquisition was controlled by the NIS-Elements Advanced Research software (Nikon Instruments).

Raw phase contrast and epifluorescence images were processed using custom scripts implemented in the MATLAB environment available at https://github.com/dibbelab/Cycloop. Briefly, the images in the red channel were used to identify single cell nuclei visible thanks to the nuclear fluorescence marker (Htb2-mCherry). Next, the nuclei centroids were used as seeds for a Voronoi tessellation to generate a single-cell region mask to crop the phase contrast images around each cell. The resulting phase contrast cropped images were used to generate a binary mask defining the region of a single cell. We applied the function *regionprops* to all the single-cell mask to quantify the radius of each cell. Fluorescence intensities were then quantified by processing the yellow fluorescence images with the binary single-cell masks described above, using the function *regionprops*. Specifically, for the non-cycling strain the fluorescence is quantified as the average fluorescence intensity in the region selected by the binary mask; while for the cycling strain the fluorescence is quantified as the maximum fluorescence intensity detected in the region selected by the binary mask. Fluorescence intensities are measured in arbitrary units (a. u.). Single-cell traces were tracked in real-time using a custom tracking algorithm previously described[19]. Briefly, the custom tracking algorithm tracks nuclei centroids (i.e., the cells) in two consecutive phase contrast images by solving an optimisation problem. The optimisation problem aims to minimising the displacements among the nuclei centroids of two consecutive images. To improve the single-cell trace datasets, we devised an offline implementation of the tracking module that performs also a reverse tracking of the cell population, which means that the tracking module was run a second time starting however from the last time frame towards the first frame. Thus, we obtained a reverse single-cell traces' dataset. Then, the algorithm combined the forward and the reverse datasets to improve the single-cell trace dataset.

**Microfluidics experiments**. The microfluidics experimental platform was initialised as previously described[30]. Briefly, a microfluidics device was filled with double distilled water for removing air bubbles. Next, two syringes filled with −MET and +MET media were first mounted on the automated actuation system and connected to the inlet ports of the microfluidics device. Afterwards, three syringes filled with double distilled water were connected to the remaining ports of the microfluidics device for balancing the flow pressure inside the device. Finally, the microfluidics device was placed in the opaque cage incubator of the microscope that was preheated to either 30 °C (nominal condition) or 27 °C (temperature perturbed condition). For microfluidics experiments performed in nominal

conditions, cells from a frozen glycerol stock (−80 °C) were resuspended in 10 mL of either methionine-free (non-cycling strain) or methionine-supplemented (cycling strain) growth medium, grown overnight in a shaking incubator at 220 r.p. m. and 30 °C, and then injected in the microfluidics device by pouring the batch culture in a syringe that was temporarily connected to the loading port of the microfluidics device. Unless otherwise specified, cells were left to settle in the chamber for 15 min fed with either methionine-depleted (non-cycling strain) or methionine-rich (cycling strain) growth medium. After that, the operator run the image acquisition and the custom MATLAB software. At the beginning of the experiment, a region of interest (ROI) was selected on the first acquired phase contrast image. Specifically, the ROI defines the area containing the *S. cerevisiae* cells that have to be segmented and tracked, and whose fluorescence signals have to be quantified. For the carbon source perturbation experiments, cells were treated as in the nominal conditions except that galactose was the only carbon source added to the growth media. For the temperature perturbation experiments, cells were grown as in the nominal conditions except that the temperature was maintained at 27 °C in lieu of 30 °C.

**Modelling**. We constructed a deterministic agent-based mathematical model to quantitatively describe the collective dynamical behaviour of cell cycle in both strains (cycling and non-cycling). Our agent-based model is based on previously published models[42–44], where each agent represents a cell in the population. The model of a single agent is based on a set of two state-dependent ordinary differential equations (ODEs), which track the evolution of cell cycle phase $\vartheta$ and cell volume $V$ in each cell:

$$\frac{\mathrm{d}}{\mathrm{d}t}\vartheta = \begin{cases} 0 & \text{if } 0 \le V < V_c \\ f(\vartheta) + z(\vartheta)u & \text{if } V \ge V_c \end{cases}, \quad (1)$$

$$\frac{\mathrm{d}}{\mathrm{d}t}V = g(\vartheta), \quad (2)$$

where $\vartheta \in \mathbb{S}^1$ is the $2\pi$-periodic cell cycle phase on the unit circle, $V \in \mathbb{R}_+$ is the cell volume, $V_c \in \mathbb{R}_+$ is the critical volume and $u \in \{0, 1\}$ is the external trigger input. The critical volume defines the minimum volume required to start the cell cycle, and it is also used to discern between mother ($V \ge V_c$) and daughter ($0 \le V < V_c$) cells. The external binary input represents the methionine-rich ($u = 0$) or methionine-depleted ($u = 1$) growth medium. The phase-dependent switching function $g : \mathbb{S}^1 \to \mathbb{R}_+$ defines the cell volume growth rate during the $G_1$ phase:

$$g(\vartheta) = \begin{cases} \beta V & \text{if } 0 \le \vartheta < \vartheta_{G_1/S} \\ 0 & \text{if } \vartheta_{G_1/S} \le \vartheta < 2\pi \end{cases}, \quad (3)$$

where $\beta \in \mathbb{R}_+$ is the volume growth rate and $\vartheta_{G_1/S}$ is the cell cycle phase value at the $G_1$ to S transition. We assumed that cell's volume grows exponentially only during the $G_1$ phase, thus neglecting the mass generated during the other phases (i.e. S-$G_2$-M), most of which is transferred to the growing bud[31]. The phase-dependent switching function $f : \mathbb{S}^1 \to \mathbb{R}_+$ models the phase oscillator dynamics. The function $f$ changes according to the specific strain. For the non-cycling strain, the function $f := f_{nc}$ is state-dependent:

$$f_{nc}(\vartheta) = \begin{cases} 0 & \text{if } 0 \le \vartheta < \vartheta_{G_1/S} \\ \omega & \text{if } \vartheta_{G_1/S} \le \vartheta < 2\pi \end{cases}, \quad (4)$$

where $\omega = \frac{2\pi}{T}$ is the angular velocity depending on the cell cycle period $T$. For the cycling strain, the function $f := f_c$ becomes state-independent:

$$f_c(\vartheta) = \omega. \quad (5)$$

Finally, the phase response curve $z : \mathbb{S}^1 \to \mathbb{R}_+$ models the linear response of the cell cycle phase $\vartheta$ to the input $u$:

$$z(\vartheta) = \begin{cases} \omega_z & \text{if } 0 \le \vartheta < \vartheta_{G_1/S} \\ 0 & \text{if } \vartheta_{G_1/S} \le \vartheta < 2\pi \end{cases}, \quad (6)$$

where $\omega_z \in \mathbb{R}_+$ is the angular velocity added to the cell cycle phase dynamics when the cell is fed with methionine-free medium.

Our agent-based model also considers cell division events. Indeed, when a cell passes through the M to $G_1$ transition ($\vartheta_{M/G_1} = 2\pi$) then a new cell (i.e. a new agent) is added to the model. The initial condition of the daughter cell depends on the state of the mother cell. Specifically, the initial phase is set to $\vartheta_0 = 0$, while the initial volume is set to $V_0 = 0.61 V_{M/G_1}$, where $V_{M/G_1}$ is the volume of the mother cell at the division time (i.e. at the M/$G_1$ transition). Therefore, the number of cells in our agent-based model is an increasing value.

The parameter values used in the agent model of the non-cycling strain are $V_c = 1$ (critical volume), $\beta = 0.0083 \text{min}^{-1}$ (volume growth rate; see ref. [31]), $T = 75$ min (nominal cell cycle period; see ref. [36]), $\omega_z = \frac{2\pi}{T}$ (angular velocity in methionine-free medium), and $\vartheta_{G_1/S} = \frac{\pi}{2}$ (phase value at $G_1$/S transition; see ref. [45]). The parameter values used in the agent model of the cycling strain are: $V_c = 1$ (critical volume), $\beta = 0.0083 \text{ min}^{-1}$ (volume growth rate; see ref. [31]), $T = 105$ min (nominal cell cycle period; see Phase estimation and budding index section), $\omega_z = \frac{2\pi}{T}$ (additional angular velocity in methionine-free medium), and

$\vartheta_{G_1/S} = \frac{4}{5}\pi$ (phase value at G1/S transition; see Phase estimation and budding index section). All simulations were performed using an initial population of $N_0 = 3$ homogeneous cells. Initial phases were uniformly spaced in the interval $[0, 2\pi]$. Initial volumes were set equal to the critical volume $V_c$. The agent-based system was solved using the MATLAB *ode15s* solver. All plots were generated in MATLAB. The code to run all the simulations and the model is available at https://github.com/dibbelab/Cycloop/tree/main/Simulator.

**Phase estimation and budding index.** For the non-cycling strain, the cell cycle phase $\vartheta \in [0, 2\pi]$ was estimated by comparing the single-cell *CLN2-YFP* trace with a periodic reference signal. The periodic reference signal CLN2$_{ref}$ was constructed according to the dynamical expression of the essential G1 cycling gene *CLN2* in the non-cycling strain:

$$\text{CLN2}_{ref}(t) = \begin{cases} 0 & \text{if } kT \leq t < kT + T_{flat} \\ \frac{1}{2} - \frac{1}{2}\cos\left(\frac{2\pi}{T_0} \times t\right) & \text{if } kT + T_{flat} \leq t < (k+1)T \end{cases}, \forall k \in \mathbb{N}_0, \quad (7)$$

where $T = T_{flat} + T_0$ is the period of the reference signal, $T_{flat}$ is a fake time interval in which the cell is assumed to be halted in the G1 phase and $T_0$ is the nominal cell cycle period in the non-cycling strain. The first part of the periodic reference signal describes the situation in which the cell has a flat fluorescence signal, that is when it is halted in the G1 phase, and thus the duration $T_{flat}$ must be set equal to the length of the compared fluorescence signal. Instead, the second part models the oscillatory *CLN2-YFP* expression in a cell that is cycling, i.e. when it is fed with methionine-free medium. At each sampling time $t$, the measured fluorescence signal was cross-correlated with the periodic reference signal evaluating the Pearson's correlation coefficient $r$. Specifically, the last part (duration equals to $T_{flat}$) of the measured fluorescence signal CLN2$_{[t-T_{flat}, t]}$ was cross-correlated with the periodic reference signal using a shifting time window in the interval $[\tau - T_{flat}, \tau]$, where $\tau \in [T_{flat}, T]$. The time point $\tau$ at which $r$ reached the maximum value was used as a time-reference to estimate the cell cycle phase $\hat{\vartheta}$ using the linear relationship:

$$\hat{\vartheta} = \frac{2\pi}{T_0} \times (\tau - T_{flat}) + \frac{\pi}{2}, \quad (8)$$

where $\hat{\vartheta} = 0$ means that the cell is at the M/G1 transition. Pearson's correlation coefficient was computed using the MATLAB function *corr*. The nominal cell-cycle period $T_0$ was set to 75 min, i.e. the duration of the cell cycle period in the non-cycling strain[36]; while the time period $T_{flat}$ was set to 30 min, according to the length of the measured fluorescence signal used to evaluate the cross-correlation. The length of the G1 phase was set to 25% of the nominal cell cycle period[45].

For the cycling strain, the cell cycle phase $\vartheta \in [0, 2\pi]$ was estimated from the single-cell *CDC10-YFP* trace using a custom procedure. The single-cell fluorescence signal was firstly binarized according to the dynamical expression of the septine protein Cdc10. Indeed, the septine protein Cdc10 contributes to form the ring at the interface between the mother and the daughter cell (i.e. the bud)[46]. Thus, the fluorescence signal is linked to the bud formation and so is detected only during the budded phase (i.e. S-G2-M phases). In the ideal case, the fluorescence signal is binary (on/off). To binarize the single-cell fluorescence trace at the sampling time $t$, the raw Cdc10-YFP signal was coarsely smoothed with a moving average filter with window size of 5 samples and then filtered again with a finer 3 samples window filter. Then, the filtered fluorescence signal was transformed in a binary signal using the threshold value $l(t)$:

$$f(x(t)) = \begin{cases} 0 & \text{if } x(t) < l(t) \\ 1 & \text{if } x(t) \geq l(t) \end{cases}, \quad (9)$$

where $x$ is the filtered fluorescence value, and $l$ is the threshold value; both evaluated at time $t$. $f(x) = 0$ means that the cell is in the off state, i.e. the unbudded phase; while $f(x) = 1$ means that the cell is in the on state, i.e. the budded phase. The threshold value $l$ at time $t$ was computed from the raw fluorescence data extracted from the final 100 min of the acquired single-cell fluorescence signals. Specifically, the raw fluorescence values for all the cells were aggregated together in a single dataset, and then a mixture of two normal distributions was fitted on the histogram of this dataset. The threshold level was chosen as the intersection point of the two normal distributions. Next, the single-cell binary signal was checked to detect the last binary transition. This can be an on to off (on → off) transition, corresponding to the start of the unbudded phase; or an off to on (off → on) transition, corresponding to the start of the budded phase. Assuming that the last transition occurred at time $t - T_t$, the estimated cell cycle phase $\hat{\vartheta}$ was computed as

$$\hat{\vartheta}(t) = \begin{cases} \frac{2\pi}{T_0} \times (t - T_t) & \text{if on → off} \\ \frac{2\pi}{T_0} \times (t - T_t) + \vartheta_{G_1/S} & \text{if off → on} \end{cases}, \quad (10)$$

where $T_0$ is the nominal cell cycle period in the cycling strain, $T_t$ is the time interval from the last binary transition to the current time and $\vartheta_{G_1/S}$ is the phase value at which occurs the G1/S transition. If a cell did not show a binary transition during the observation period, then the cell cycle phase could not be estimated. To further improve the quality of results, we set a minimum duration of 14 min between two successive binary transitions. Part of this procedure was used to

estimate the nominal cell cycle period $T_0$ and the phase value $\vartheta_{G_1/S}$ from the raw fluorescence traces of an uncontrolled +MET experiment (Supplementary Data 1). Specifically, the single-cell raw fluorescence signals were binarized using a threshold value $l$. Then, the nominal duration of the unbudded phase $T_{G_1} = 42$ min was computed as the median of the intervals measured in all binary traces between an on to off (on → off) and an off to on (off → on) transition. Similarly, the nominal duration of the budded phase $T_{S-G_2-M} = 63$ min was computed as the median of the intervals measured in all binary traces between an off to on (off → on) and an on to off (on → off) transition. Finally, the numerical values of $T_0$ and $\vartheta_{G_1/S}$ were obtained as

$$T_0 = T_{G_1} + T_{S-G_2-M} = 105 \text{ min}, \quad (11)$$

$$\vartheta_{G_1/S} = 2\pi \times \frac{T_{G_1}}{T_{G_1} + T_{S-G_2-M}} = \frac{4}{5}\pi. \quad (12)$$

The budding index (B.I.) was computed as the percentage of budded cells in the population:

$$\text{B.I.}(t) = \frac{N_{S-G2-M}(t)}{N(t)} \times 100, \quad (13)$$

where $N(t)$ is the total number of cells at time $t$ and $N_{S-G2-M}(t)$ is the number of budded cells (i.e. cells in the S-G2-M phases) at time $t$.

**Quantification of synchronisation.** To quantify the degree of cell cycle synchronisation across the cell population, we computed two synchronisation indices: a) the mean coherence phase $R$ of the Kuramoto order parameter[47] and b) the amplitude Power of the leading peak in the power spectrum of the average YFP fluorescence signal.

The mean phase coherence $R \in [0, 1]$ and the mean phase $\psi \in [0, 2\pi]$ are defined as the magnitude and argument of the Kuramoto order parameter, respectively[47]:

$$R(t)e^{j\psi(t)} = \frac{1}{N(t)} \sum_{m=1}^{N(t)} e^{j\vartheta_m(t)}, \quad (14)$$

where $N(t)$ is the time-varying number of cells, while $\vartheta_m(t)$ is the phase of the $m$th cell, both evaluated at time $t$. When $R$ is equal to 1, all cells are synchronised. Conversely, when $R$ is equal to 0, cells are totally desynchronised.

The amplitude Power of the leading peak is computed from the power spectral density (PSD) of the normalised YFP fluorescence signal. Specifically, the mean of the average YFP signal is subtracted from the signal itself, then the MATLAB function *pmcov* of the *Signal Processing Toolbox* is used to compute the parametric estimation of the PSD. The leading peak of the PSD is found with the MATLAB function *max*. The leading peak in the PSD is associated to a specific period. We decided to discard a peak value associated to a period longer than the duration of the YFP signal. In this case, we classified that peak as not determined (N.D.).

**Control algorithms.** All control algorithms were devised to change the control input (i.e., the growth medium) once a new system output (i.e., the estimated cell cycle phases) was available. Moreover, the control input was held constant between two consecutive phase measurement times (zero-order hold method[48]). All control algorithms were implemented through custom scripts in the MATLAB environment. The illustrative code for each control algorithm is available at https://github.com/dibbelab/Cycloop/tree/main/Illustrative_Platform_Code.

*Stop&go control.* The stop&go control algorithm is an event-triggered feedback control strategy that was devised to synchronising the cell cycle across a population of non-cycling cells[42]. At each sampling time $t_k$, the stop&go algorithm computes the percentage of cells in the G1 phase by means of the estimated cell cycle phases (see Phase estimation and budding index section). If the percentage of cells in the G1 phase is higher than a fixed threshold $\nu_\%$, then the algorithm delivers a −MET pulse of duration $D_{-Met}$ to the cells, otherwise cells are kept in +MET medium. Controller's parameters used in our implementation were: $\nu_\% = 50\%$ (threshold value) and $D_{-Met} = 30$ min(duration of −MET pulse). Note that $\nu_\% = 100\%$ in the ideal implementation illustrated in Fig. 3a.

*Reference oscillator.* The reference oscillator strategy is a state feedback control strategy that was devised to synchronise the cell cycle across a population of cells. We adapted the control strategy proposed by Bai and Wen[38] to design a state feedback control law able to steer the cell cycle phases to converge towards a reference phase. Here, the reference phase evolves over time according to the dynamics of a virtual phase oscillator that interacts with all the cells in the population through a virtual star-like coupling. We constructed the virtual reference oscillator through an ODE model that describes the evolution of the reference oscillator phase $\vartheta_r$ on the unit circle $\mathbb{S}^1$ over time:

$$\frac{d}{dt}\vartheta_r = \omega_r + \gamma \sum_{m=1}^{N} \sin(\vartheta_m - \vartheta_r), \quad (15)$$

where $\omega_r \in \mathbb{R}_+$ is the natural frequency of the reference oscillator, $\gamma \in \mathbb{R}_+$ is the coupling strength, $N \in \mathbb{N}$ is the number of cells and $\vartheta_m \in \mathbb{S}^1$ is the cell cycle phase of the $m$th cell in the population. Note that the number of cells changes over time at each sampling time $t_k$ (see Phase estimation and budding index section). Moreover, the cell cycle phases were held constant between two consecutive phase measurement times (zero-order hold method[48]). At each sampling time $t_k$, the algorithm first measures the cell cycle phases through the phase estimation algorithm (see Phase estimation and budding index section). Let $N_k$ be the number of cells whose phases have been measured at the sampling time $t_k$. Then, the algorithm computes the control action $u(t_k)$ to apply to the cell population in the interval $[t_k, t_{k+1}]$ through the state feedback control law:

$$u(t_k) = \begin{cases} u_{OFF} = ' + MET', & \text{if } \sum_{m=1}^{N_k} a_m(t_k)\sin(\vartheta_m(t_k) - \vartheta_r(t_k)) \geq 0 \\ u_{ON} = ' - MET', & \text{if } \sum_{m=1}^{N_k} a_m(t_k)\sin(\vartheta_m(t_k) - \vartheta_r(t_k)) < 0 \end{cases}, \quad (16)$$

where $\vartheta_r(t_k)$ is the phase of the reference oscillator at the sampling time $t_k$, $\vartheta_m(t_k)$ is the cell cycle phase of the $m$th cell measured at the sampling time $t_k$; and $a_m(t_k) \in \{0, 1\}$ is a time-dependent coefficient associated to the $m$th cell denoting if that cell is in the G1 phase at the sampling time $t_k$, that is:

$$a_m(t_k) = \begin{cases} 1 & \text{if } 0 \leq \vartheta_m(t_k) < \vartheta_{G_1/S} \\ 0 & \text{if } \vartheta_{G_1/S} \leq \vartheta_m(t_k) < 2\pi \end{cases}, \quad (17)$$

where $\vartheta_{G_1/S}$ is the phase at the G1 to S transition according to the phase estimation algorithm (see Phase estimation and budding index). Finally, the algorithm integrates the ODE model of the reference oscillator to obtain the value of the reference phase $\vartheta_r$ at the next sampling time $t_{k+1}$. To integrate the reference oscillator ODE, we used the MATLAB numerical solver *ode45*. Controller's parameters used in our implementation were: $\omega_r$ equals to the nominal natural velocity of the cell cycle in the cycling strain (see Modelling section), and $\gamma = 1$ (coupling strength). The initial condition of the reference oscillator phase was $\vartheta_{r,0} = 0$. For the complete details of the reference oscillator strategy, including the derivation of the state feedback control law, refer to the Supplementary Notes.

**Reporting summary**. Further information on research design is available in the Nature Research Reporting Summary linked to this article.

## Data availability
The authors declare that the data supporting the findings of this study are available within the paper and its Supplementary Information files. Source data for Figs. 2–4 and Supplementary Figs. 1–9 are provided at https://github.com/dibbelab/Cycloop[49]. The microscopy data have been deposited in Zenodo, https://doi.org/10.5281/zenodo.4516319. Strains and plasmids used in this study are available from the corresponding author upon reasonable request.

## Code availability
The source code that supports the findings of this study is available from GitHub at https://github.com/dibbelab/Cycloop[49].

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

## Acknowledgements

The authors would like to thank Sahand Jamal Rahi (EPFL, Switzerland) for the SJR14a4d *S. cerevisiae* strain (i.e. the non-cycling strain), Gilles Charvin (IGBMC, France) for the GC84-35B *S. cerevisiae* strain (i.e. background of the cycling strain), Cathal Wilson (TIGEM, Italy) for the insightful help with strain construction and for the plasmid pRS41N-GAP-CYC, Ciro Talotti (TIGEM, Italy) for help with media preparation, and Jeff Hasty (UCSD, USA) for the master mould of the microfluidic device. This work was supported by the Fondazione Telethon grant TGM16SB1 and by the COSY-BIO (Control Engineering of Biological Systems for Reliable Synthetic Biology Applications) project, which has received funding from the European Union's Horizon 2020 research and innovation programme under grant agreement 766840.

## Author contributions

G.P., S.N. and D.d.B. designed the research; G.P. developed and supervised the controllers' designs, modelling and experiments; S.N. performed the experiments and developed image analysis and phase estimation algorithms; F.G. and A.L.R. equally contributed to the research; F.G. helped to design the controller strategies and the phase estimation algorithms; A.L.R. helped to perform experiments, image analysis and modelling; D.F. helped to develop the stop&go controller and the cell cycle model; T.G. engineered the cycling yeast strain; M.d.B. helped in controllers' design and modelling; G.P., S.N. and D.d.B. analysed the data; D.d.B. proposed the concept and supervised the project; G.P., S.N. and D.d.B. wrote the manuscript.

## Competing interests

The authors declare no competing interests.
