## [Peer Review File · Nature Communications]

Reviewers' Comments:

Reviewer #1:

Remarks to the Author:

The manuscript by Perrino et al describes a technique to synchronize yeast cell cycle using a combination of microscopy, microfluidics and real-time control. As it builds on some of the most recent technical advancements in in vivo control, this paper certainly offers an elegant, yet compelling and robust demonstration of how closed-loop feedback control can be applied to steer the dynamics of complex biological systems. To the best of this reviewer's knowledge, this is the first demonstration of this kind for an endogenous system as complex as *S. cerevisiae*'s cell cycle. Interestingly, this point is briefly mentioned by the authors but not elaborated much further than an "en passant" statement: I would encourage Perrino and colleagues to provide their readers with a broader context of the literature so that the intrinsic technical value of this work can be fully appreciated.

While I have no doubt around such technical value, I am struggling to assess to what extent this approach advances the body of knowledge the scientific community has around cell cycle. The manuscript seems to suggest that the novelty envisioned by the author is mainly centered on the method: how impactful their discovery ultimately is, will depend on the ability of their method to overcome the limitations of current approaches to synchronize cells. Assessing this impact is virtually impossible for a non-expert reader and, in my opinion, this ends up being an unnecessary gap in the manuscript. The authors only hint at the limitations of current phase-synchronisation methods: what are they? How does their method fill such gaps? What are the implications for practitioners, e.g. industrialists, if any? None of these questions is considered in the manuscript; they all should. In general, the paper would benefit from a more systematic review of the literature on cell cycle control (and exogenous control of genetic circuits, in general) that makes the reader appreciate the real depth/implications of the results reported.

So much for my fundamental concerns with the paper. Minor comments/questions follow:

- The rationale for using the "non-cycling" strain only becomes clear after both the circuits have been introduced/described. The authors should consider reshuffling the content so that the reason for using both circuits is clear from the get-go;
- I note that "cyber-yeast" is a term that other authors have previously used to refer to similar but different setups. To avoid any confusion, I believe that the authors should consider adopting a different name for their work;
- Fig. 2: why is fluorescence in 2b not constant? Is there any effect of cell stress while yeast is loaded in the microfluidic device (early stage of the experiment)?
- Fig. 2: why is cell count going down periodically in the Open Loop experiments (e.f. from 400 min onwards in 2f)?
- Fig. 4: control performance seems to degrade before the interruption of the control action: why is that the case? Have the authors carried out a negative control (i.e. kept the controller going beyond the time threshold used)? What happens in that case?

Reviewer #2:

Remarks to the Author:

In this manuscript, the authors propose a cyber-genetic feedback control mechanism to synchronize yeast cells to the same phase of the cell cycle. The authors propose to synchronize a population of genetically engineered yeast cells by interfacing them with a computer that controls a microfluidics device. In their setup, the computer calculates the appropriate feedback control action, which consists of adding or removing the amino acid methionine to and from the growth media of the yeast cells. The authors use two yeast strains that have been genetically engineered to initiate the cell cycle when methionine is removed from their growth media. The non-cycling

yeast strain lacks cyclins CLN1-3 and has an exogenous cyclin gene CLN2 under the control of a methionine-repressible promoter; thus, this strain only cycles in the absence of methionine. The cycling yeast strain has an extra copy of the cyclin gene CLN2 under the control of the same methionine-repressible promoter; thus, this strain cycles continuously.

To achieve synchronization across the yeast cell population, the authors propose and implement a bang-bang feedback control strategy for the non-cycling yeast strain and a state feedback control strategy (reference oscillator) for the cycling yeast strain. First, the authors propose and demonstrate the implementation of a stop&go (bang-bang) feedback control strategy for the non-cycling yeast strain. The feedback controller's action results in synchronization across the non-cycling yeast population for up to 1000 minutes. Second, the authors propose and demonstrate a state feedback control strategy (reference oscillator) for the cycling yeast strain that synchronizes yeast population for up to 600 mins. The authors also propose a model predictive controller for the cycling yeast strain, but it fails, possibly due to the unmodeled dynamics in the underlying mathematical model of the yeast population.

The results of the paper are clearly illustrated and the data are qualitatively compelling. But a number of number of major issues should be highlighted:

1- Motivation for impact of work is weak. The authors state: "In addition to providing a new avenue for yeast cell cycle synchronisation, our work shows that computers can automatically steer complex biological processes towards desired behaviours similarly to what is currently done with robots and autonomous vehicles." However, this has already been shown in a number of previous publications spanning almost a decade (e.g. control of ERK signaling, etc...)

The authors' second motivation is to enable improved measurements of transcriptional, proteomic, metabolomic, and signaling readouts across a synchronized yeast cell population. This point needs to be either illustrated or argued and unpacked better. Is the vision there is that an mRNA-seq experiment will be coupled to such an elaborate setup? Feasibility and ease of adoption need to be explained if this is indeed the motivation. It is the case that genomics experiments that proceed at the single cell level can now back-calculate cell cycle state and hence correct for that in biological conclusions. In this case, what would be the impact of the work?

2- Quantification of performance is missing—figures and text rely on qualitative statements of "worse", "best", etc... with no statistical or quantitative metrics for comparison for performance of different strategies. This is a case that can be made for every trace shown in the whole manuscript, but it is especially salient for last figure where a reference oscillator cell actually exists, but we have no idea, quantitatively, how far or close the experimental behavior is to that reference. In the same vein, robustness aside, how does open-loop periodic pulsing compare to closed-loop? It is puzzling that it seems to behave great despite all the fluctuations that happen in cells (even in absence of huge wholesale shifts such as in temperature or carbon source).

3- With argument about robustness as a driving motivation for closed loop control, it seems necessary to check whether the feedback controller for the cycling yeast strain (one with tracking of reference oscillator) is robust to changes in temperature and media (analogous to supplementary Figs 4 and 5). In effect, robustness should be checked for perturbation that happen in the middle of closed loop operation (e.g. media flow or heat shock in middle of experiment) and perhaps for perturbations disturb in real time cell cycle dynamics, such as salt stress, that immediately arrests cell cycle, etc..

4- The purpose of the MPC experiment is rather unclear, since no conclusion is really drawn from those data. For example, it is unclear why the mathematical model of yeast cells is simplified before constructing the model predictive controller (lines 558-566). The authors motivate this simplification by stating that it reduces the computational complexity of the model, but it could also be the underlying cause of the MPC controller's failure. I would suggest removing the MPC control strategy altogether from the paper or moving it to supplementary materials, unless there is a point that I might have missed here, and which needs to be made more clearly.

Other comments:

- The reference oscillator controller sometimes turns on and off for what seems like only a few minutes at a time (orange vertical lines in Fig 4, panels o, t, and Supplementary Fig 8). Is this the feedback controller overly sensitive? Does it have much of an effect when it removes or adds methionine for a few mins at a time? These quick control actions seem to somewhat correspond to small changes in the budding index. Is there something happening in the yeast cell population at that time?
- What is the vertical dotted line in Supplementary Figure 5, panels a-e? Does it signify that the temperature perturbation occurs at 550 mins?
- When the galactose perturbation is performed on line 203, is the open loop controller recalibrated to the new periodicity? If the cell cycle is slower, T_u and duration should be different.
- Paper has a number of biological imprecise statements, such as in reference to cycling strain "the cell cycle can also be initiated on demand by inducing exogenous CLN2 expression via methionine removal". Not clear what this means as the cell cycle is proceeding anyway uninterrupted, so by definition, continuously "initiated"
- Paper has a number of unclear characterizations, as one example "This experimental characterisation of the non-cycling yeast strain thus confirmed that it behaved as expected." Unclear what "expected" means in this case. Since the strain still has to have Cln1/3 deleted, "expected" could be what was reported in its original publication, or what is expected in wild type strain. Also, how does the strain background affect the ability to generalize for control of WT cell cycle dynamics?

Point-by-point reply to REVIEWERS' COMMENTS

In what follows we replied to all the reviewers' comments and suggestions. We renumbered the comments to make cross-reference easier and we formatted in *italic* the reviewers' comments, whereas our replies are formatted as normal text. We also highlighted changes made in the revised manuscript to make it easier for the reviewers to detect them. The revised manuscript now includes three additional experiments and additional analyses. Consequently, we have now revised two figures (Fig. 4 and Supplementary Fig. 4) and included two additional supplementary figures (Supplementary Fig. 7 and Supplementary Fig. 9).

Reviewer #1 (Remarks to the Author):

1.1 The manuscript by Perrino et al describes a technique to synchronize yeast cell cycle using a combination of microscopy, microfluidics and real-time control. As it builds on some of the most recent technical advancements in in vivo control, this paper certainly offers an elegant, yet compelling and robust demonstration of how closed-loop feedback control can be applied to steer the dynamics of complex biological systems. To the best of this reviewer's knowledge, this is the first demonstration of this kind for an endogenous system as complex as S. cerevisiae's cell cycle. Interestingly, this point is briefly mentioned by the authors but not elaborated much further than an "en passant" statement: I would encourage Perrino and colleagues to provide their readers with a broader context of the literature so that the intrinsic technical value of this work can be fully appreciated.

We thank the reviewer for the encouraging comments and suggestions. We have now added a new paragraph to the Introduction section (lines 94-99) to briefly review the literature on the application of Control Engineering to biological processes, and also to the Discussion section (lines 348-352) to emphasise the technical novelty of our work.

1.2 While I have no doubt around such technical value, I am struggling to assess to what extent this approach advances the body of knowledge the scientific community has around cell cycle. The manuscript seems to suggest that the novelty envisioned by the author is mainly centered on the method: how impactful their discovery ultimately is, will depend on the ability of their method to overcome the limitations of current approaches to synchronize cells. Assessing this impact is virtually impossible for a non-expert reader and, in my opinion, this ends up being an unnecessary gap in the manuscript. The authors only hint at the limitations of current phase-synchronisation methods: what are they? How does their method fill such gaps? What are the implications for practitioners, e.g. industrialists, if any? None of these questions is considered in the manuscript; they all should. In general, the paper would benefit from a more systematic review of the literature on cell cycle control (and exogenous control of genetic circuits, in general) that makes the reader appreciate the real depth/implications of the results reported.

Again, we appreciate the comments of the reviewer and suggestions. We have now expanded the Introduction section (lines 58-82) to include a broader background on the topic of yeast cell cycle synchronisation and how our approach differs from the rest. We also added a paragraph to the Discussion section to highlight the potential applications of this technology (lines 356-362).

1.3 Minor comments/questions follow:

- The rationale for using the "non-cycling" strain only becomes clear after both the circuits have been introduced/described. The authors should consider reshuffling the content so that the reason for using both circuits is clear from the get-go;

Thanks for the suggestion, we have now moved the text explaining the challenges of controlling a cycling strain at the beginning of the Results section (lines 136-143).

- I note that "cyber-yeast" is a term that other authors have previously used to refer to similar but different setups. To avoid any confusion, I believe that the authors should consider adopting a different name for their work;

Duly noted, we have now removed this term from the manuscript.

- Fig. 2: why is fluorescence in 2b not constant? Is there any effect of cell stress while yeast is loaded in the microfluidic device (early stage of the experiment)?

The reviewer is right, this could be caused by an initial stress caused by the cell loading process. Although we cannot be sure this is the cause, we have now computed the average fluorescence of the constitutively expressed red nuclear protein in this experiment and we found it to be highly correlated to the green fluorescence, thus suggesting this to be a global effect on protein expression caused by some form of stress or adaptation to the microfluidics environment.

- Fig. 2: why is cell count going down periodically in the Open Loop experiments (e.f. from 400 min onwards in 2f)?

We believe this to be caused by the fact that cells reach their maximum number in the chamber (about 1000) and then when they divide their number increases but cells at the edge of the chamber are

pushed out of the field of view all together thus causing the drop in the number of cells (this is apparent in the videos – Supplementary Movies 2 and 3).

- Fig. 4: control performance seems to degrade before the interruption of the control action: why is that the case? Have the authors carried out a negative control (i.e. kept the controller going beyond the time threshold used)? What happens in that case?

The reviewer is right, as time goes by there is a degradation in the phase estimation algorithm due to a drop in performance of the image segmentation step caused by cell crowding in the chamber. The degradation on phase estimation in turn causes a decrease in the controller's performance.

Reviewer #2 (Remarks to the Author):

The results of the paper are clearly illustrated and the data are qualitatively compelling. But a number of number of major issues should be highlighted.

2.1 - Motivation for impact of work is weak. *The authors state: "In addition to providing a new avenue for yeast cell cycle synchronisation, our work shows that computers can automatically steer complex biological processes towards desired behaviours similarly to what is currently done with robots and autonomous vehicles." However, this has already been shown in a number of previous publications spanning almost a decade (e.g. control of ERK signaling, etc...) The authors' second motivation is to enable improved measurements of transcriptional, proteomic, metabolomic, and signaling readouts across a synchronized yeast cell population. This point needs to be either illustrated or argued and unpacked better. Is the vision there is that an mRNA-seq experiment will be coupled to such an elaborate setup? Feasibility and ease of adoption need to be explained if this is indeed the motivation. It is the case that genomics experiments that proceed at the single cell level can now back-calculate cell cycle state and hence correct for that in biological conclusions. In this case, what would be the impact of the work?*

We thank the reviewer for the constructive comments. Indeed, similar comments were also raised by Reviewer 1 (point 1.1). As correctly pointed out by the reviewer, the motivations of our work were both technical and scientific. Regarding the technical motivation, although it is correct that in the last 10 years applications of control engineering to biological systems have been described, so far these applications have only been achieved for "simple" processes such as the regulation of gene expression from an inducible promoter, or regulation of protein localisation. To the best of our knowledge, this is the first demonstration of computer-based feedback control of an endogenous system as complex as *S. cerevisiae*'s cell cycle, as also stated by Reviewer 1 in her/his comments (point 1.1). However, we acknowledge that we did not do a good job in placing our work in the context of the recent literature; therefore, we have now expanded the Introduction section to illustrate the recent progresses of biocontrol engineering and the originality of our contribution (lines 94-99). Regarding the scientific motivation, we have now illustrated in the Introduction section (lines 58-82) the many approaches proposed over more than 20 years to synchronise yeast cell cultures and explained how our closed-loop feedback strategy is different from the rest. Finally, our statement regarding measurements of -omics data was perhaps misleading; indeed, we just wanted to point out that yeast biologists have strived for years to synchronise yeast populations to better study the cell cycle and to have more robust measurements of morphological, physiological and molecular quantities. We have now rephrased this statement in the Introduction to make this point clearer (lines 53-57). Finally, we added a paragraph to the Discussion section to highlight the potential applications of our technology (lines 356-362).

2.2 Quantification of performance is missing—figures and text rely on qualitative statements of "worse", "best", etc.... with no statistical or quantitative metrics for comparison for performance of different strategies. *This is a case that can be made for every trace shown in the whole manuscript, but it is especially salient for last figure where a reference oscillator cell actually exists, but we have no idea, quantitatively, how far or close the experimental behavior is to that reference. In the same vein, robustness aside, how does open-loop periodic pulsing compare to closed-loop? It is puzzling that it seems to behave great despite all the fluctuations that happen in cells (even in absence of huge wholesale shifts such as in temperature or carbon source).*

We apologise for not having satisfactorily described in our previous version of the manuscript the quantitative analyses that indeed we had performed to compare open- and closed-loop experiments. Specifically, for each experiment we did measure two metrics to quantify the synchronisation and reported them in Supplementary Figure 4 (which we have now revised to include the three additional experiments we performed for this revision). The two metrics are: (1) the time-average of the mean phase coherence R ; and (2) the amplitude ($Power$) and the period of the leading peak of the power spectrum of the mean fluorescence signal. We have now better described these metrics (lines 197-201) and reported their values in the main text where relevant. For the cycling strain, we have also added a new supplementary figure (Supplementary Fig. 9) to compare the phase of the reference oscillator to the mean phase of the cell population, as suggested by the reviewer.

Regarding the open-loop performances, these can be easily compared to closed-loop performances in Supplementary Fig. 4, where we reported all the quantitative metrics for open- and closed-loop control strategies for both the non-cycling and cycling strains. It can be appreciated that in nominal growth conditions, the closed-loop is always better or on par with the open-loop control, however the “correct” period and duration of the stimulation must be carefully chosen for the open loop strategy to work. Indeed, when growth conditions are perturbed by changing temperature or carbon source, the open loop strategy fails both for the non-cycling and cycling strains, unless the period and duration are properly adjusted, which can be done only by a trial-and-error approach (refer to Fig. 2 and Supplementary Fig. 3). On the other hand, closed loop control is able to automatically find the correct stimulation to apply without any prior knowledge or human intervention.

2.3 *With argument about robustness as a driving motivation for closed loop control, it seems necessary to check whether the feedback controller for the cycling yeast strain (one with tracking of reference oscillator) is robust to changes in temperature and media (analogous to supplementary Figs 4 and 5). In effect, robustness should be checked for perturbation that happen in the middle of closed loop operation (e.g. media flow or heat shock in middle of experiment) and perhaps for perturbations disturb in real time cell cycle dynamics, such as salt stress, that immediately arrests cell cycle, etc..*

Following the reviewer’s suggestion, we have now performed three additional experiments with the cycling strain to assess the robustness of the closed-loop feedback strategy and to compare it to the open-loop strategy. We could not perform a perturbation in the middle of the closed-loop experiment as our microfluidics device has only two inputs, which we need to deliver the control inputs (medium with or without methionine), whereas we would need a third input to apply the disturbance. However, we performed two additional open-loop experiments with the same period of 80 min in nominal growth conditions (revised Supplementary Fig. 7k-o) and in galactose (revised Fig. 4k-o) and a new closed-loop experiment in galactose (revised Fig. 4p-t). We chose the galactose perturbation because it is one of those conditions in which state-of-the-art synchronisation methods fail to work; as the cell cycle slows down, the difference between mother and daughter cells is enhanced thus causing desynchronization after only one cycle. It can be appreciated both visually ((revised Fig. 4k-t) and quantitatively (revised Supplementary Fig. 4) that whereas the open-loop strategy performs well in nominal conditions, in perturbed conditions it fails to achieve synchronisation, whereas the closed-loop strategy is still able to automatically synchronise the cell population. We have now described these new results in the Result section (lines 274-332).

2.4 *The purpose of the MPC experiment is rather unclear, since no conclusion is really drawn from those data. For example, it is unclear why the mathematical model of yeast cells is simplified before constructing the model predictive controller (lines 558-566). The authors motivate this simplification by stating that it reduces the computational complexity of the model, but it could also be the underlying cause of the MPC controller's failure. I would suggest removing the MPC control strategy altogether from the paper or moving it to supplementary materials, unless there is a point that I might have missed here, and which needs to be made more clearly.*

The reviewer is right, chronologically we performed MPC first and then we searched for alternative control strategies as the MPC was not satisfactory, and came out with the reference oscillator controller, which worked very well. As we spent a lot of time on the MPC strategy, we could not bring ourselves to remove it, but now, after reading the reviewer's comment, we decided to remove the MPC completely from this manuscript, as it does not add much to our results, as correctly pointed out by the reviewer.

2.5 *Other comments:*

- The reference oscillator controller sometimes turns on and off for what seems like only a few minutes at a time (orange vertical lines in Fig 4, panels o, t, and Supplementary Fig 8). Is this the feedback controller overly sensitive? Does it have much of an effect when it removes or adds methionine for a few mins at a time? These quick control actions seem to somewhat correspond to small changes in the budding index. Is there something happening in the yeast cell population at that time?

Indeed, the controller decides whether to deliver -MET or +MET at each sampling time $T_s = 2$ min. Hence it may happen that between two consecutive sampling times, the controller decides to switch the growth medium, thus giving rise to these spikes. Increasing the sampling time would make the control input less variable but could also decrease the controller's performance, therefore we decided to keep the sampling time short, also considering that cells do not "sense" such quick changes in the methionine concentration and that methionine is a natural amino-acid, and as such it does not cause any stress to the cells. The reviewer is correct in noticing that the budding index is somehow correlated to the control input. Indeed, the reference oscillator decides whether to provide methionine-rich or methionine-poor medium to the cells, on the basis of the estimated cell cycle phase in each cell. Sometimes, error in the phase estimation algorithm, or biological noise in the reporter protein expression, can give rise to noisy phase estimates that cause brief spikes in the control input.

- What is the vertical dotted line in Supplementary Figure 5, panels a-e? Does it signify that the temperature perturbation occurs at 550 mins?

Thanks for noticing this mistake. We have removed both vertical lines in Supplementary Figure 5a-e, indeed the temperature was shifted for the whole experiment.

- When the galactose perturbation is performed on line 203, is the open loop controller recalibrated to the new periodicity? If the cell cycle is slower, T_u and duration should be different.

Indeed, the reviewer is right, T_u and duration should be different for the open loop control, but we did not recalibrate them for this experiment as the correct combination is not known and multiple trial-and-error experiments would be required to find it. Indeed, the point of this experiment was to show that the open loop control requires careful selection of period and duration, which cannot remain constant but must be adjusted for each specific growth condition; on the contrary, the closed-loop strategy can automatically find and adjust in real-time the correct combination to achieve synchronisation. We have now detailed the period and duration used in the open loop experiments in the main text to make it clearer to the reader (lines 207-214).

- Paper has a number of biological imprecise statements, such as in reference to cycling strain "the cell cycle can also be initiated on demand by inducing exogeneous CLN2 expression via methionine removal". Not clear what this means as the cell cycle is proceeding anyway uninterrupted, so by definition, continuously "initiated"

We thank the reviewer for pointing this out. We have now carefully revised the manuscript to remove imprecisions and oversimplifications.

- Paper has a number of unclear characterizations, as one example "This experimental characterisation of the non-cycling yeast strain thus confirmed that it behaved as expected." Unclear what "expected" means in this case. Since the strain still has to have Cln1/3 deleted, "expected" could be what was reported in its original publication, or what is expected in wild type strain. Also, how does the strain background affect the ability to generalize for control of WT cell cycle dynamics?

We have now revised the manuscript and clarified the text where necessary. Regarding the generalisation of the approach, our method requires cell cycle division mutants to work as we need a way to induce the transition from G1 to the S phase. We used two different strains (non-cycling and cycling) to demonstrate the generality of the approach to different cell cycle mutants, and because the cycling strain is closer to the wild type as only Cln3 is deleted in this case. To make this point clearer we have now moved the text explaining the challenges of controlling a cycling strain versus the non-cycling strain at the beginning of the Results section (lines 136-143).

Reviewers' Comments:

Reviewer #1 (Remarks to the Author)

The authors have now addressed all my concerns and I am satisfied with their answers. Both the additional qualitative and quantitative points contribute to strengthen the message and broaden the reach of these results.

I am persuaded that these findings can now be considered material contributions to both eukaryotic cell cycle studies and the nascent area of "in vivo control of biomolecular networks".

Reviewer #2 (Remarks to the Author)

The authors in general did a good job answering my queries, especially with respect to quantification of performance through inclusion of Supplementary Fig. 9. Furthermore, the authors tried to situate their work in the literature and provide motivation. However, I think that the paper still needs to be more honest about previous work that uses feedback controllers in biological systems in two ways. First, authors need to strictly refer to computer-based feedback control in their statements because otherwise there are feedback controllers with synthetic components that go beyond the "regulation of gene expression from an inducible promoter or the regulation of protein localization". For example, there are synthetic feedback controllers that control the growth of a cell population (You et al. in Nature, 2004). Is that less of a "complex" process than feedback control on the cell cycle, as the authors claim on lines 97-99 and in the discussion lines 349-350? Second, it is not true that computer-based feedback control "has only been applied to simple processes such as the regulation of gene expression from an inducible promoter, or to control protein localisation" (lines 97-99) and that theirs "is the first demonstration of computer-based feedback control of an endogenous system as complex as *S. Cerevisiae*'s cell cycle" (lines 349-350)? Those are pretty big, imprecise claims. Here are two references that the authors might have missed-- Neuman et al. 2015 in ELife is doing optogenetic feedback control of neurons firing and the feedback control strategy is computed by a computer.
<https://pubmed.ncbi.nlm.nih.gov/26140329/>. Also, please see <https://pubmed.ncbi.nlm.nih.gov/30340045/> for optogenetic, computer-controlled feedback control of a complex process—that of the mating pathway, also in yeast.

I also disagree with the last portion of the discussion (lines 363-365), which was added after review. I don't think that implementing the stop & go controller genetically is trivial. How would they know what feedback strategy to use? Their feedback controller turns on and off as needed because that's how it works. Is that strategy implementable genetically? Does the feedback strategy not depend on the media, temperature, other environmental conditions? I think that this paragraph is unsubstantiated.

Minor comments:

- Line 36: "The cell cycle acts as a sort of global oscillator regulating cell growth and division" -> imprecise statement, please revise
- Line 60: "underlying principle" -> "underlying principles of"
- Line 63: "chemical species" -> "chemicals"
- Lines 77-82: missing citations about self-cycling fermenters and how they work.
- Lines 97 - 99: Imprecise statement: Clarify that the authors are only referring to computer-based feedback control. Many feedback controllers have been built inside single cells and at the population level from synthetic biological parts.
- Line 118: "and shown" -> "as shown"

Point-by-point reply to REVIEWERS' COMMENTS

In what follows we replied to reviewers' comments and suggestions. We renumbered the comments to make cross-reference easier and we formatted in *italic* the reviewers' comments, whereas our replies are formatted as normal text. We also highlighted changes made in the revised manuscript to make it easier for the reviewers to detect them.

Reviewer #1 (Remarks to the Author):

The authors have now addressed all my concerns and I am satisfied with their answers. Both the additional qualitative and quantitative points contribute to strengthen the message and broaden the reach of these results. I am persuaded that these findings can now be considered material contributions to both eukaryotic cell cycle studies and the nascent area of "in vivo control of biomolecular networks".

We thank the reviewer for finding our contribution of interest and worth of publication.

Reviewer #2 (Remarks to the Author):

The authors in general did a good job answering my queries, especially with respect to quantification of performance through inclusion of Supplementary Fig. 9. Furthermore, the authors tried to situate their work in the literature and provide motivation. However, I think that the paper still needs to be more honest about previous work that uses feedback controllers in biological systems in two ways.

We thank the reviewer for appreciating our efforts and finding our manuscript much improved.

2.1 *First, authors need to strictly refer to computer-based feedback control in their statements because otherwise there are feedback controllers with synthetic components that go beyond the "regulation of gene expression from an inducible promoter or the regulation of protein localization". For example, there are synthetic feedback controllers that control the growth of a cell population (You et al. in Nature, 2004). Is that less of a "complex" process than feedback control on the cell cycle, as the authors claim on lines 97-99 and in the discussion lines 349-350?*

Point taken. Indeed several examples of synthetic circuits exist that modify the behaviour of the population. We have now modified the text to restrict the field to computer-based feedback control and also revised our claims (please refer to the response to point 2.2).

2.2 *Second, it is not true that computer-based feedback control "has only been applied to simple processes such as the regulation of gene expression from an inducible promoter, or to control protein localisation" (lines 97-99) and that theirs " is the first demonstration of computer-based feedback control of an endogenous system as complex as S. Cerevisiae's cell cycle" (lines 349-350)? Those are pretty big, imprecise claims. Here are two references that the authors might have missed-- Neuman et al. 2015 in ELife is doing optogenetic feedback control of neurons firing and the feedback control strategy is computed by a computer. <https://pubmed.ncbi.nlm.nih.gov/26140329/>. Also, please see <https://pubmed.ncbi.nlm.nih.gov/30340045/> for optogenetic, computer-controlled feedback control of a complex process—that of the mating pathway, also in yeast.*

We agree with the reviewer that our claims were not entirely correct. We have now referenced the manuscripts suggested by the reviewers and downplayed our claims in the light of the recent literature.

2.3 I also disagree with the last portion of the discussion (lines 363-365), which was added after review. I don't think that implementing the stop & go controller genetically is trivial. How would they know what feedback strategy to use? Their feedback controller turns on and off as needed because that's how it works. Is that strategy implementable genetically? Does the feedback strategy not depend on the media, temperature, other environmental conditions? I think that this paragraph is unsubstantiated.

We understand that this paragraph may confuse readers. Indeed, we did not mean to imply that building such an embedded controller would be easy, but rather that our work on computer-based feedback control suggests that if built, such an "embedded" stop & go controller may work. However, to avoid any confusion, and as suggested by the reviewer, we have now decided to remove this paragraph from the discussion.

Minor comments:

- Line 36: "The cell cycle acts as a sort of global oscillator regulating cell growth and division" -> imprecise statement, please revise
- Line 60: "underlying principle" -> "underlying principles of"
- Line 63: "chemical species" -> "chemicals"
- Lines 77-82: missing citations about self-cycling fermenters and how they work.
- Lines 97 - 99: Imprecise statement: Clarify that the authors are only referring to computer-based feedback control. Many feedback controllers have been built inside single cells and at the population level from synthetic biological parts.
- Line 118: "and shown" -> "as shown"

We modified the text according to the suggested edits.